# Cold Agglutinin Disease: Improved Understanding of Pathogenesis Helps Define Targets for Therapy

Sigbjørn Berentsen [1,*], Shirley D'Sa [2], Ulla Randen [3,4], Agnieszka Małecka [5,6] and Josephine M. I. Vos [7,8]

1   Department of Research and Innovation, Haugesund Hospital, Helse Fonna Hospital Trust, NO-5504 Haugesund, Norway
2   UCLH Centre for Waldenström and Associated Conditions, University College London Hospitals NHS Foundation Trust, London NW1 2BU, UK
3   Department of Pathology, Akershus University Hospital, NO-1478 Nordbyhagen, Norway
4   Institute of Clinical Medicine, University of Oslo, NO-0318 Oslo, Norway
5   Department of Haematology and Department of Pathology, Oslo University Hospital, NO-0318 Oslo, Norway
6   KG Jebsen Centre for B-cell Malignancies and Institute of Clinical Medicine, University of Oslo, NO-0318 Oslo, Norway
7   Department of Hematology, Amsterdam University Medical Centers, University of Amsterdam, 1105 AZ Amsterdam, The Netherlands
8   Lymphoma and Myeloma Center Amsterdam, Sanquin, 1105 AZ Amsterdam, The Netherlands
*   Correspondence: sigbjorn.berentsen@haugnett.no

**Abstract:** The last 2 decades have seen great progress in understanding the pathogenesis of cold agglutinin disease (CAD) and development of effective therapies. Cold agglutinins can cause hemolytic anemia as well as peripheral circulatory symptoms such as acrocyanosis. We distinguish CAD, a well-defined clinicopathologic entity, from secondary cold agglutinin syndrome. This review addresses the histopathologic, immune phenotypic, and molecular features that allow CAD to be classified as a distinct clonal lymphoproliferative disorder of the bone marrow, recently recognized in the WHO classification. We discuss recent data on the possible overlap or distinction between CAD and Waldenström's macroglobulinemia. Two major steps in the pathogenesis of CAD are identified: clonal B-cell lymphoproliferation (leading to monoclonal IgM production) and complement-mediated hemolysis. Each of these steps constitutes a target for treatment. Established as well as novel and experimental therapies are reviewed.

**Keywords:** cold agglutinin disease; autoimmune hemolytic anemia; lymphoproliferative; monoclonal gammopathy; Waldenström's macroglobulinemia; complement; therapy

## 1. Introduction

Cold agglutinin disease (CAD) is a rare hemolytic disorder that accounts for 15–30% of autoimmune hemolytic anemias (AIHA) [1–6]. The autoantibodies that mediate CAD are designated as cold agglutinins (CA), referring to their ability to agglutinate erythrocytes at temperatures below central body temperature. CA were discovered in 1903 and related to hemolysis in 1937 [7,8]. Systematic descriptions of CAD appeared in the 1950s–60s [9–11], whereas major progress in understanding the pathogenesis and treatment has been achieved during the last 2–3 decades [1,12,13].

Recent literature and international consensus distinguish CAD from CA syndrome (CAS) [1,5,12,14]. While CAD (often termed primary CAD) is a well-defined clinicopathologic entity as will be described below, CAS is a similar but heterogeneous group of CA-mediated AIHAs that can occur secondarily to other clinical disease, such as specific infections (*Mycoplasma pneumoniae* pneumonia, Epstein–Barr virus infection, SARS-CoV-2 infection, and others) and malignancies (in particular B-cell lymphoma) [5,15–17].

In CAD, a clonal lymphoproliferative disorder (LPD) of the bone marrow, further described below, results in production of CA [18]. Binding of circulating CA to cell surface antigens results in agglutination of erythrocytes and complement-mediated hemolysis, which explain the respective clinical features of CAD [10,19,20]. Thus, two major pathogenetic steps are identified, bone marrow clonal lymphoproliferation and complement-dependent hemolysis, each of which constitutes a logical and feasible target for therapy [18–21].

## 2. Properties of Cold Agglutinins

CA are produced by B-lymphocytes in the bone marrow [18,21] and to some extent by mature plasma cells [22–24]. CA in CAD are monoclonal immunoglobulin M (IgM) antibodies or, in rare cases, IgG [18,25–27]. In fact, the first monoclonal protein ever identified was a CA from a patient with CAD [28]. In an observational study of 232 patients with CAD, sufficient data on immunoglobulins were available in 222 patients [26]. Monoclonal IgM was found in 91% of patients in whom a clonal immunoglobulin could be detected; IgG in 4.5%, and both IgM and IgG in 2.8%. Two patients had monoclonal IgA, which was, however, probably not identical with the CA [26,29,30]. In 20% of all patients, a monoclonal protein had not been detected by standard methods [26]. The light chain restriction was κ in 90.4% of patients with a monoclonal band. Most CA in CAD have specificity for erythrocyte surface carbohydrate antigen I (big I), which also has been identified on leukocytes and platelets [31–34]. While neutrophil aggregates and pseudo-neutropenia have been observed in rare cases of CAD [32,35], low platelet counts have not been reported. Rarely, the antigen specificity can be anti-i (little i), anti-Pr, or anti-IH [31,36,37].

In the majority of cases, anti-I CA genes have a highly conserved structure, consisting of *IGHV4-34* together with *IGKV3-20* or the very similar *IGKV3-15* [38], which suggests that both the heavy and light chain are important for the disease activity. The *IGHV4-34* gene is required to encode anti-I specific CA [13,38–40]. This gene appears to be overrepresented among the coding unit repertoire, although it accounts for a very small fraction of normal circulating immunoglobulins [41].

Framework region 1 of the *IGHV4-34* gene, encoding a $Gln^6$-$Trp^7$ (QW) and $Ala^{24}$-$Val^{25}$-$Tyr^{26}$ amino acid sequence, is essential for recognition of the I antigen and is not mutated in CAD [38,42,43]. Two other regions of this gene also appear to be important for binding to the I antigen. The first one is the N-glycosylation sequon, which is located in complementarity determining region (CDR) 2 within the antigen-binding pocket; therefore, inactivating mutations preventing attachment of glycans likely modulate binding [44]. Second, a mutation hotspot within framework region 3 with a $Lys^{90}$-$Leu^{91}$-$Ser^{92}$ (KLS) amino acid sequence is located close to the antigen binding pocket. Both regions are mutated in most CAD patients. The number of mutations negatively correlates with hemoglobin levels, as mutations might enable better antigen binding [38]. The affinity and specificity for I antigen binding also depends on the heavy-chain $CDR_H3$ hypervariable region and the light-chain variable region [38,42]. *IGKV3-20* and *IGKV3-15* are very often used (59% and 15%, respectively), which also seems to contribute to I antigen binding [38,45–48]. Furthermore, a low level of mutations in the *IGKV* CDR3 region, as well as the *IGKV3-20* gene, has been shown to correlate with younger age at diagnosis [38].

The term CA is derived from the biological properties of these autoantibodies, not from the relation between clinical manifestations and ambient temperatures [19,49]. CA have highest affinity for their antigen at 0–4 °C but can also react at higher temperatures within their individual thermal amplitude, defined as the highest temperature at which agglutination occurs [19,50]. In general, the pathogenicity of a CA depends on the antigen specificity, thermal amplitude, and clonality, monoclonal CA being more pathogenic than polyclonal CA [15,50,51]. There is also some evidence that the degree of IgM polymerization can influence the pathogenicity, as hexameric IgM has been shown to be more pathogenic than the usual, pentameric form [52,53]. Hexameric IgM has been implicated as playing a pathogenetic role in CAD, although no systematic studies have been published [54]. The

activity of CA in plasma or serum at a given temperature is semiquantitatively expressed by the titer, defined by the highest dilution at which agglutination can be seen [10,19].

Because of this variability in biologic activity, there is no fixed correlation between IgM levels and CA titers when samples from different patients are compared [26,55]. The thermal amplitude has been shown to be more important than the titer for the pathogenic potential [50]. If the thermal amplitude exceeds 28–30 °C, antigen binding will occur at temperatures normally found in the acral part of the circulation, and the CA will be pathogenic. In some patients, the thermal amplitude can approach 37 °C [55].

CA are found in a proportion of healthy people and patients with unrelated diseases, although reported percentages are highly variable [55,56]. Such normally occurring CAs are present in titers usually below 64 and nearly always below 256, have low thermal amplitude, and are polyclonal [21,55]. Typically, they are incidentally detected in individuals without hemolysis, and these persons do not have CAD or CAS.

### 3. CAD as a Clonal Lymphoproliferative Disorder

A low-grade, clonal B-cell LPD affecting the bone marrow has recurrently been observed in patients with CAD, although in some cases these changes are difficult to detect [18,57]. Until the last decade, this LPD was variably classified as lymphoplasmacytic lymphoma (LPL), marginal zone lymphoma (MZL), small lymphocytic lymphoma (identical with the bone marrow disorder seen in chronic lymphocytic leukemia), unclassified low-grade LPD, monoclonal gammopathy, or reactive lymphocytosis [26,58,59]. In 2014, however, Randen and coworkers published a systematic study of bone marrow histopathology in 54 patients with typical CAD in whom they found a surprisingly uniform disorder, which they termed CA-associated LPD [18]. Recently, CAD has been recognized in the World Health Organization Classification of Hematolymphoid Neoplasms and The International Consensus Classification of Mature Lymphoid Neoplasms as a distinct entity among mature B-cell neoplasms [60,61]. This neoplastic disorder is not considered as being a malignant lymphoma [60,61]. Transformation to diffuse large B-cell lymphoma occurred in only 3.4% of the cases during 8 years in a series of 232 patients with CAD [26].

In the histopathology study, 40/54 bone marrow biopsies showed small lymphoid cells, mainly within nodular aggregates that were often tiny, but also as a sparse interstitial infiltration (Figure 1) [18,60]. In 14/54 patients, scattered clonal lymphocytes were the only finding. Mature plasma cells accounted for <5% of the nucleated cells and were seen surrounding the aggregates and sparsely scattered in the parenchyma, but not within the aggregates. No paratrabecular growth pattern, mast cell infiltration, or fibrosis could be identified. The lymphoid cells expressed the B-cell markers CD19, CD20, CD22, PAX5, CD79a, CD79b, IgM, and monotypic light chain, most often κ. CD5 was expressed in 40% of the cases. The cells were negative for BCL6, MUM1, CD23, and cyclin D1. Plasma cells were positive for IgM and, usually, κ light chain. Flow cytometric studies have found a ratio between κ- and λ-positive B-cells > 3.5 in bone marrow aspirates from 90% of patients [57,58].

Trisomy 3 was present in all bone marrow samples in a series of 15 CAD patients, often in combination with trisomy 12 or 18 [62–64]. Trisomy 18 seemed to be associated with the poorest response to therapy, while patients having only trisomy 3 had the best response. Exome sequencing of clonal B-cells showed a relatively low mutational burden; however, several recurrent mutations demonstrated in CAD are also found in lymphoma [65]. *KMT2D* was shown to be mutated in almost 70% of patients, while *CARD11* was mutated in over 30%. *IGLL5* and *CXCR4* were mutated in 61% and 28% of patients, respectively [65,66]. *CARD11* and *CXCR4* mutations corresponded with hemoglobin levels [65]. Several gene mutations that have been observed affect the NF-κB pathway as well as chromatin modification or organization. Of high interest, the *MYD88* L265P mutation is not (or rarely) displayed in CAD [18,38,65,67].

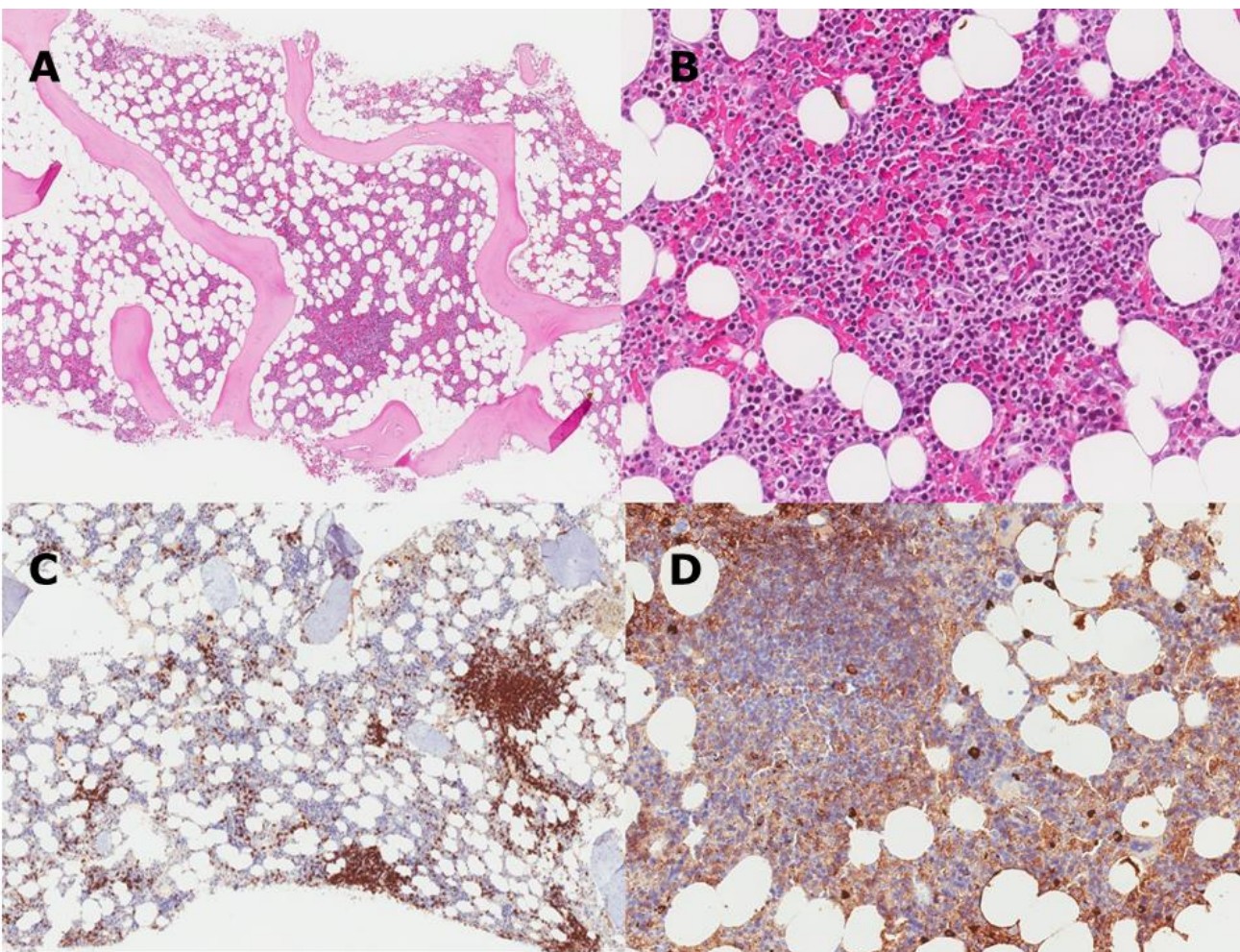

**Figure 1.** Cold agglutinin-associated lymphoproliferative bone marrow disorder. Panel (**A**): Nodular lymphoid aggregate (HE-staining, 50×). Panel (**B**): Aggregate of small lymphoid cells with round to oval nuclei and scant cytoplasm (HE-staining, 280×). Panel (**C**): CD20 staining shows nodular as well as interstitial infiltration of B-cells (50×). Panel (**D**): IgM-positive B-cells in the infiltrate and scattered IgM-positive plasma cells around the infiltrate and diffusely throughout the bone marrow (145×).

In clinical practice, there will be a variable proportion of patients in whom a clonal LPD is not demonstrated. However, a retrospective, descriptive study in which 176 CAD patients had available data on bone marrow biopsies showed that the detection rate of CA-associated LPD increased from 23% to 68% on centralized biopsy review by expert pathologists familiar with this disorder (Figure 2) [26]. Another retrospective study evaluated 20 patients with cold-antibody AIHA, among whom 4 had no evidence of a clonal disorder on serum electrophoresis and routine bone marrow examination [68]. In 3 of these, however, the authors demonstrated a clonal rearrangement of immunoglobulin heavy and/or light chains. These findings support the assumption that all patients with CAD have a clonal LPD.

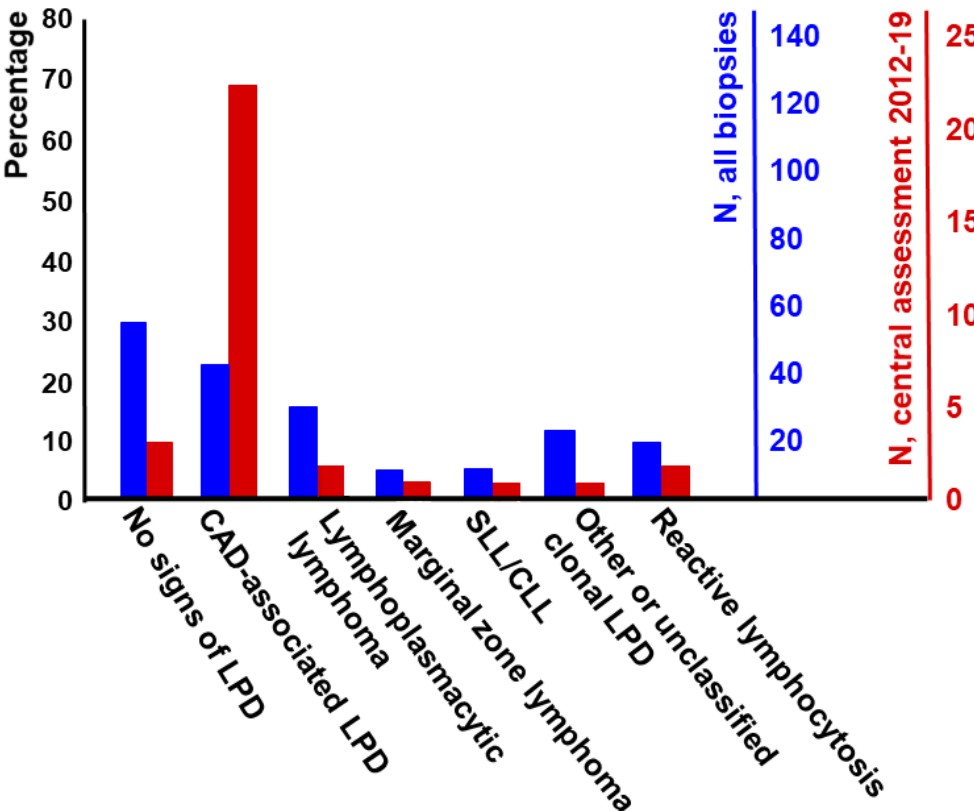

**Figure 2.** Bone marrow biopsy findings in 176 CAD patients with available histology data; comparison by percentage. Centralized revision increased the detection rate of clonal LPD and the percentage of biopsies interpreted as CAD-associated LPD. Blue bars, all biopsies. Red bars, biopsies that underwent a centralized review from 2012 through 2019. CLL, chronic lymphocytic leukemia; LPD, lymphoproliferative disorder; SLL, small lymphocytic B-cell lymphoma. Figure first published in Blood 2020 [26], reused under general permission. © the American Society of Hematology.

## 4. CAD and Waldenström's Macroglobulinemia: Distinction or Overlap?

Waldenström's macroglobulinemia (WM) is defined by the combination of monoclonal IgM in serum at any concentration and LPL in the bone marrow with at least 10% infiltration [69]. Pathogenic CA can be found in patients diagnosed with WM or monoclonal gammopathy of clinical significance (MGCS) [70–72]. In a series of 122 untreated patients with WM, 3% had CA-mediated hemolytic anemia [73]. Among 172 subjects with monoclonal IgM in serum, high-titer CA was detected in 8.5% [51]. In patients with CAD, the associated bone marrow changes were previously classified as LPL in a variable percentage [58,59], which often led to the conclusion that CAD represents a spectrum of clonal LPD overlapping with LPL/WM [70,74,75].

However, the 2014 systematic study of bone marrow changes in CAD revealed, even with regard to histomorphologic findings, differences between CAD and WM (Table 1) [18,60,76]. For example, while an increased number of mast cells are readily identified in trephine biopsy sections from patients with WM, this feature is typically not seen in CAD [18,60,76]. Overall, the histomorphologic and immune phenotypic features in CAD seem more similar to MZL than LPL [18]. Strikingly, and already mentioned, the *MYD88* L265P mutation, shown to be present in the vast majority of patients with LPL/WM, is usually not detected in CAD [18,66,67,77,78]. There are also differences between CAD and WM in the heavy chain gene use, while, as far as we know, the light chain gene use has not been elucidated in WM [38,39].

**Table 1.** Bone marrow histopathologic features in CAD and LPL/WM.

| Feature | CAD | LPL |
| --- | --- | --- |
| Growth pattern | Small nodules, no diffuse infiltrates or intrasinusoidal growth | Nodular and/or diffuse |
| Location | Intertrabecular infiltrates | Paratrabecular/intertrabecular infiltrates |
| Clonal cells | Small B-cells and usually plasma cells | Small B-cells, lymphoplasmacytoid cells, plasma cells |
| Plasma cells | Widely spread in the parenchyma | Admixed in the infiltrates |
| Mast cells | No mast cell infiltration | Mast cell infiltration present |

Based on data from Randen et al. [18], Naresh et al. [60], and Owen et al. [76].

The large 2020 multinational observational study of CAD confirmed that a significant number of patients had been diagnosed as having LPL on bone marrow biopsies examined by unselected pathology laboratories [26]. When re-examined in pathology departments with special interest in CAD, however, this proportion decreased from 17% to 5%, while the proportion of biopsies diagnosed with CA-associated LPD increased from 23% to 68% (Figure 2). In our experience, most CAD patients originally interpreted as having LPL/WM are *MYD88* L265P negative.

These observations allow the conclusion that, in general, CAD and WM are distinct entities. If a patient with CAD is diagnosed as having "*MYD88* L265P negative LPL", the histopathologic diagnosis should be reconsidered. On the other hand, we do see occasional patients with otherwise typical CAD and bone marrow changes convincingly demonstrating features of LPL, including the *MYD88* L265P mutation [5,26,67]. It may be a matter of definition or individual additional characteristics whether these cases should be classified as CAD with atypical histopathology or CAS secondary to LPL/WM [5].

**5. CAD as a Complement-Mediated Hemolytic Anemia**

In the acral parts of the circulation, cooling of the blood allows binding of CA to the antigen, resulting in agglutination of erythrocytes and fixation of complement C1 complex (C1qrs) to antigen-bound IgM (Figure 3). This binding initiates the classical complement activation pathway by activating C1s, a serine protease that splits C4 and C2, followed by the formation of C3 convertase (C4b2a) and subsequent cleavage of C3 [20,79–81]. This step leaves the involved erythrocytes opsonized with the split product C3b [79,80]. C3b-labelled erythrocytes are prone to phagocytosis by the mononuclear phagocytic system in the liver, a process known as extravascular hemolysis [79–82]. On the surviving erythrocytes, C3b is further degraded to C3d, which is not recognized by the phagocytes but allows the opsonized cells to be identified in the laboratory by the direct antiglobulin test (DAT) [5,83,84]. C3b can also react with the split products C2a and C4b to form C5 convertase, which cleaves C5. This step initiates the terminal complement cascade, which leads to formation of the C5b-9 complex (membrane attack complex) on the cell surface and intravascular hemolysis. Due to intact CD55- and CD59-mediated regulation in CAD, however, terminal complement activity is thought to be limited or absent except in severely affected patients and acute exacerbations [79,85]. Hemoglobinuria, a specific marker of intravascular hemolysis, was found in 15% in a retrospective series of 89 patients with CAD [59].

It is unknown whether the same complement-dependent mechanisms apply to the rare cases of IgG-mediated CAD, as IgG is a weaker complement activator than IgM. Of the IgG subclasses, IgG3 is the most potent activator; IgG1 is a weak activator; and IgG2 is a still weaker one, while IgG4 does hardly activate complement [86,87]. Furthermore, IgG-mediated CAD appears different from IgM-mediated disease in some clinical aspects, such as the therapeutic effect of splenectomy and, possibly, corticosteroids [27].

Most patients with CAD have low levels of C4 because of continuous consumption, which presumably acts as a rate-limiting factor for further complement activation in stable disease [26,58,85]. When an acute phase reaction occurs, the production of complement proteins is enhanced; C4 is replete, and exacerbation of hemolysis may ensue [20,58,85,88].

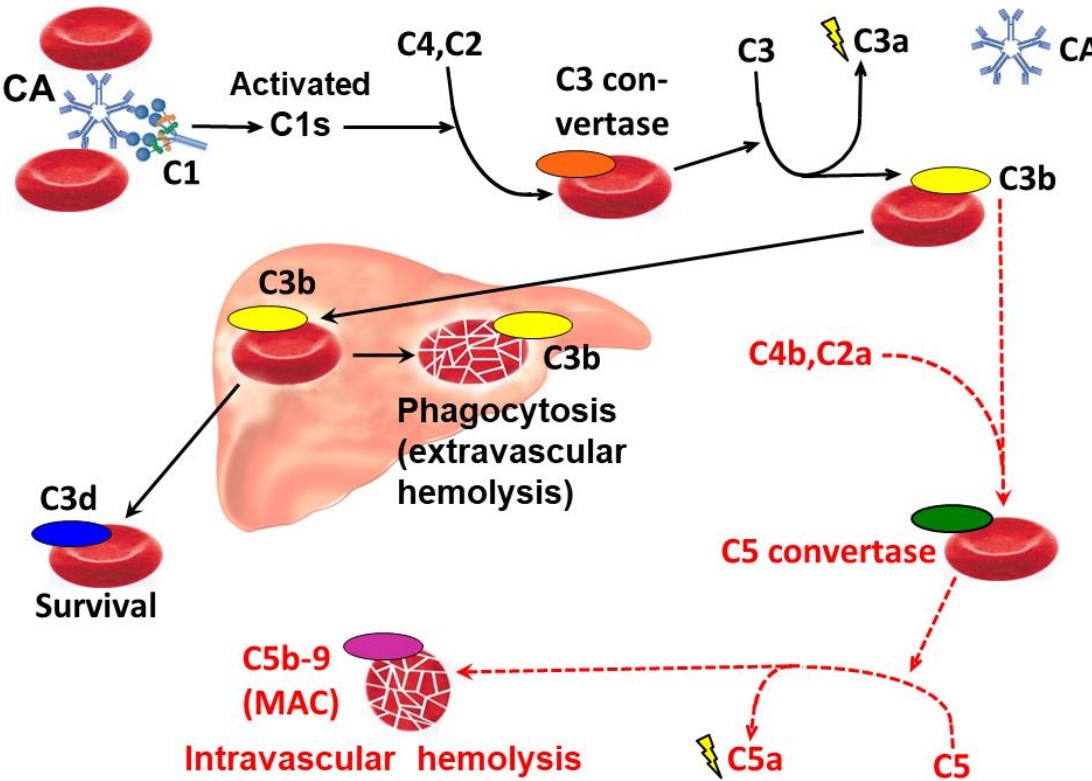

**Figure 3.** Complement-mediated hemolysis in CAD. Only relevant pathways are shown. On the surface of erythrocytes agglutinated by IgM cold agglutinin (CA), the antigen–antibody complex binds complement protein complex C1, which initiates the classical activation pathway. On warming to central body temperature, CA detaches from the cells. Sequential cleavage of C4, C2, and C3 leaves the erythrocytes coated with C3b, which makes them prone to extravascular hemolysis in the liver. C3b may also react to form C5 convertase, which triggers the terminal complement cascade, resulting in formation of the membrane attack complex (MAC) and intravascular hemolysis. Lightning symbols denote anaphylotoxin properties of soluble split products. Continuous black lines indicate major pathways. Dashed red lines indicate minor pathways active in severe disease.

## 6. Epidemiology and Clinical Features

In a recent, large descriptive study of unselected patients with verified CAD, the mean age was 76 years, while the mean age at disease onset was reported to be 67 years (lower range, 32 years) and the mean age at diagnosis was 68 years [26]. The male-to-female ratio was 0.56. The epidemiology of patients diagnosed with CAD shows large variations with geography and climate. In Norway, the prevalence was estimated to be 20 cases per million and the incidence to be 1.9 cases per million per year, while the corresponding figures for Lombardy, Northern Italy, were 5 cases per million and 0.5 cases per million per year, respectively [26]. Although seasonal variations have been documented [89–91], hemolysis is present all year round in patients with CAD [49].

In the same study, the mean hemoglobin (Hb) level at presentation was 9.3 g/dL (median, 9.2; range, 4.5 g/dL–15.3 g/dL) [26]. As shown in Figure 4, 88% of the patients had hemolytic anemia whereas 12% had compensated hemolysis [21,26]. The anemia was classified as mild (Hb 10 g/dL to lower limit of normal) in 24% of the patients, moderate (Hb 8 g/dL–10 g/dL) in 37%, and severe (Hb < 8 g/dL) in 27%.

Acute exacerbation of hemolytic anemia occurs in febrile illness in 40–70% of patients with CAD [26,58] and can also be triggered by major trauma or major surgery [85,92,93]. Estimates in the literature on transfusion dependency are highly variable; the largest series of unselected patients with well-documented CAD found that 40–50% had received transfusions at some time [26,58,59].

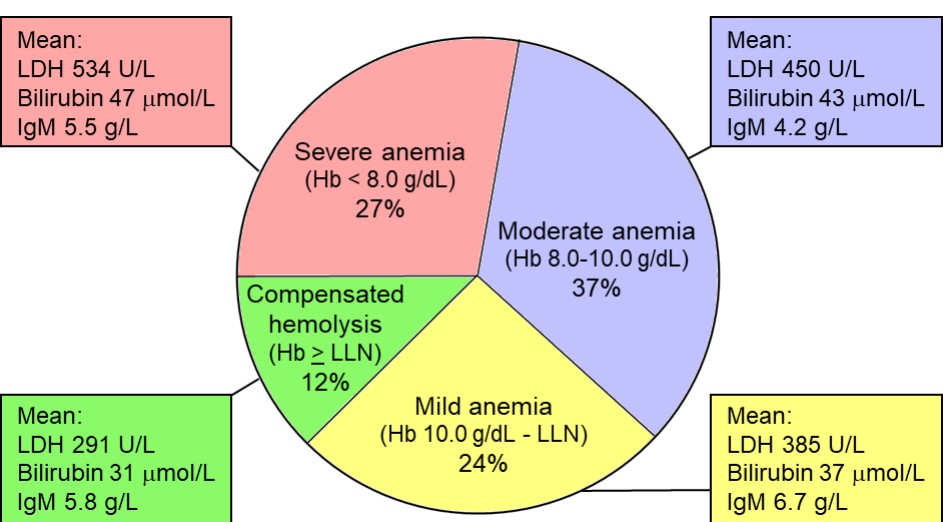

**Figure 4.** Severity of anemia in cold agglutinin disease; percentages of patients. Severity of anemia correlates nicely with markers of hemolysis (LDH and bilirubin levels), but not with IgM levels, as expected because of variable biologic activity of cold agglutinins. Hb, hemoglobin level; LDH, lactate dehydrogenase; LLN, lower limit of normal (Hb 11.5 g/dL in women and 12.5 g/dL in men). Based on data from Berentsen et al. [26]. Figure first published in Front Immunol 2019 [21], reused under a Creative Commons CC-BY license 4.0 (https://creativecommons.org/licenses/by/4.0/, accessed on 15 September 2022). © S. Berentsen.

Cold-induced, agglutination-mediated symptoms from the capillary circulation have been reported in 50–90% of the patients in systematic series, usually ranging from acrocyanosis to disabling Raynaud-like phenomena, while livedo reticularis is less common and gangrene is rare [26,58]. Any of the two latter manifestations should warrant the exclusion of cryoglobulinemia as a differential diagnosis or a concomitant property of the cold-reactive IgM [9,71].

Patients with CAD differ from one another with respect to clinical phenotype. In an observational study comprising 210 unselected patients with sufficient data, hemolytic anemia with no or mild circulatory symptoms was recorded in 69.5% of the patients, while 21% of the patients had hemolytic anemia with moderate or severe circulatory symptoms and 9.5% had circulatory symptoms with compensated hemolysis [26]. These phenotypes did not show any association with the demonstration of a clonal LPD by bone marrow histology or the response to B-cell-directed therapies. As will be explained below, however, they are expected to affect the responsiveness to complement-directed therapies [94,95].

Fatigue is common, but the exact frequency remains to be determined. In 24 symptomatic CAD patients entering a prospective trial, the Functional Assessment of Chronic Illness Therapy (FACIT) fatigue scale mean score at baseline was 32.5, which is comparable to scores seen in advanced cancer [96–98]. Probably, fatigue is not caused by the anemia only, but can also be attributed to the complement activation by itself, mediated by anaphylotoxin properties of soluble split products (Figure 3) [21,96,99].

Like other AIHAs, CAD carries a risk of thrombosis. A clinical observational study and two registry-based studies, all from Europe and the USA, confirmed a slightly increased relative risk of venous thromboembolism of 1.7–3.1 as compared with the general population or matched controls, while the risk of arterial thrombosis seemed less marked [26,100,101]. In contrast, a Japanese study confirmed the risk of thrombosis but indicated a more marked

increase in the risk of arterial versus venous thromboembolism [102]. The largest unselected studies did not demonstrate any association between risk of thrombosis and severity of anemia, but one study found a correlation with the degree of hemolysis [26,100]. Smaller studies of severely affected patients, however, may indicate a higher risk in this group [103]. Data on survival in CAD are conflicting, as some studies have found a survival similar to the general population while other studies indicate an increased mortality [26,90,101].

## 7. Diagnosis

A history or observation of typical cold-induced circulatory symptoms in an anemic patient and the finding of marked erythrocyte agglutination on blood smear microscopy should lead the clinician to suspect CAD, although these are not valid diagnostic criteria [5,6].

The criteria for CAD diagnosis are chronic hemolysis, positive DAT, monospecific DAT positive for C3d, demonstration of CA at a significant titer, and absence of overt malignancy or specific infection that might cause secondary CAS (Table 2) [1,5,14,60,84]. Detection of hemolysis is based on biochemical markers including unconjugated (or total) bilirubin, lactate dehydrogenase (LDH), and haptoglobin, while absolute reticulocytosis is a useful confirmative finding although not invariably present. As extravascular hemolysis predominates, bilirubin measurement is particularly important [5,6,96]. DAT is usually negative for IgG, but weak positivity for IgG is found in up to 20% of the cases [58,104]. It is difficult to put forward a clear cutoff for the CA titer; however, most patients will have a titer $\geq 64$ at 4 °C, often much higher [26,58,59]. Additional criteria that are not required but will be useful to confirm CAD are monoclonal IgMκ in serum or plasma (or, rarely, IgMλ or IgG), a ratio between κ- and λ-positive B-cells > 3.5 (or, rarely, <0.9) in bone marrow aspirates, demonstration of CA-associated LPD by bone marrow histopathology, and absence of the *MYD88* L265P mutation [5,60].

**Table 2.** Diagnostic criteria for CAD.

| Level | Criteria | Procedures and Comments |
|---|---|---|
| Required for diagnosis | Chronic hemolysis | As assessed by high bilirubin, low haptoglobin, high LDH, and, often, high absolute reticulocyte count |
| | Monospecific DAT strongly positive for C3d | DAT is usually negative for IgG, but occasionally weakly positive |
| | Cold agglutinin titer $\geq 64$ at 4 °C | Specimen must be kept at 37–38 °C from sampling until plasma/serum has been removed from the cells/clot |
| | No overt malignant disease or relevant infection | Clinical assessment for malignancy. Radiology as required. Exclude recent infection with *Mycoplasma* or Epstein–Barr virus |
| Confirmative criteria not required for diagnosis | Monoclonal IgMκ in plasma/serum (or, rarely, IgG or λ phenotype) | Specimen must be kept at 37–38 °C from sampling until plasma/serum has been removed from the cells/clot |
| | Ratio between κ- and λ-positive B-cells > 3.5 (or, rarely, <0.9) | Flow cytometry in bone marrow aspirate |
| | Cold agglutinin-associated lymphoproliferative disorder by histopathology | Bone marrow biopsy |
| | *MYD88* L265P mutation not found | As assessed in bone marrow |

DAT, direct antiglobulin test; LDH, lactate dehydrogenase.

To avoid false negative results and incorrect quantitative assessments, appropriate precautions must be observed in sampling and handling of specimens for CA titration, immunoglobulin class quantifications, and serum protein electrophoresis [1,5,105]. To ensure that blood samples keep at 37–38 °C from drawing to serum has been separated from the clot (or plasma from the cells), collecting tubes must be prewarmed and placed in a warming cabinet or water bath until centrifugation, and the use of a warm centrifuge is highly recommended. Serum protein electrophoresis and bone marrow biopsy are recommended in all patients [5,6,105].

## 8. Management

Patients who are not in complete remission after therapy should avoid low ambient temperatures and use warm clothing, especially protecting acral parts of the body. Even though these recommendations are largely based on theoretical considerations and clinical experience, the rationale is fairly strong [5,84,89,90]. Although many patients have discovered these measures by themselves before they see the specialist, they should be explained that the purpose is also to protect against hemolysis, not only against circulatory discomfort. It is equally important for the hematologist to provide doctors and nurses with adequate information. In the ward or outpatient department, patients with CAD should keep warm and avoid infusion of cold liquids. Any bacterial infection should be treated [1,19,85].

Pharmacological therapy is not indicated in patients with compensated hemolysis or mild anemia, provided that fatigue and cold-induced circulatory symptoms are absent or tolerable [1,5,84]. Until recently, however, the attitude towards active treatment has probably been too restrictive due to the lack of effective regimens and an underestimation of the symptom burden. Observational studies from Europe and the USA indicate that drug therapy has been attempted in 70–80% of the patients [26,58,59]. With the advent of effective therapies, patients should be offered treatment if they have symptomatic anemia, circulatory symptoms interfering with daily living, or significant fatigue [1,5].

Corticosteroids should not be used to treat CAD [1,5,12,106]. The response rate is low, probably less than 20%, although some retrospective series may indicate a higher response rate [4,26,58,59]. The few responders will often require unacceptably high doses to maintain the remission [106], and the risk of osteoporosis, skeletal events, and diabetes is hardly acceptable [5,107]. Splenectomy does not work because most of the extravascular hemolysis occurs in the liver. Exceptions have been reported among the rare cases of IgG-mediated CAD [27]. Effective therapies in CAD are directed against the pathogenic B-cell clone or, recently, against the complement system.

## 9. B-Cell-Directed Therapies

The safety and efficacy of treatment directed at the pathogenic B-cell clone has been confirmed by several studies. Two prospective, nonrandomized trials and several observational "real-life" studies showed a beneficial effect of rituximab monotherapy [26,58,108,109]. The prospective trials found partial response (PR) in approximately 50% of the patients, very few complete responses (CR), and a median response duration of 6.5–11 months (range, 2–42 months) [108,109]. Rituximab monotherapy is generally well tolerated and has become the most commonly used documented therapy for CAD [6,110,111]. Retreatment will often, but not always, induce a new response in relapsed patients [26,108]. There are no published data on rituximab maintenance in CAD.

Rituximab plus oral fludarabine resulted in a 76% response rate with 21% CR and a median increase in Hb by 3.1 g/dL (4.0 g/dL in those who achieved CR) according to a prospective trial [112]. The criteria for CR included disappearance of any histologic and flow cytometric signs of a clonal bone marrow LPD. Forty-one per cent of the participants experienced grade 3–4 neutropenia, and infections were frequent. A later follow-up study showed a median response duration of 77 months with >40% of the patients enjoying sustained response for 80–170 months but, unfortunately, also confirmed a suspected risk of late malignancies [26,112].

The combination of rituximab (375 mg/m² day 1) with bendamustine (90 mg/m² day 1–2) at a 4-week interval for 4 cycles fixed duration was investigated in a prospective, non-randomized trial published in 2017 [113]. Seventy-one per cent of patients responded; 40% achieved CR and 31% PR. Median Hb increase was 4.4 g/dL in the complete responders, 3.9 g/dL in those achieving PR, and 3.7 g/dL in the whole cohort. One-third of the patients developed temporary neutropenia grade 3–4, but only 11% had infection (including infection without neutropenia). Some patients experienced a long time to response (TTR; median 1.9 months; range, 0.25–12) and even longer time to best response (median 7 months; range, 1.5–30) [113]. Follow-up data were published in 2020 as part of a larger descriptive study, which showed a still higher response rate at 78%, including 53% CR [26]. The estimated median response duration was >88 months, and 77% of participants were still in remission after 5 years (Figure 5). The follow-up study also indicated that the bendamustine plus rituximab regimen is safe with regard to late-occurring malignancies.

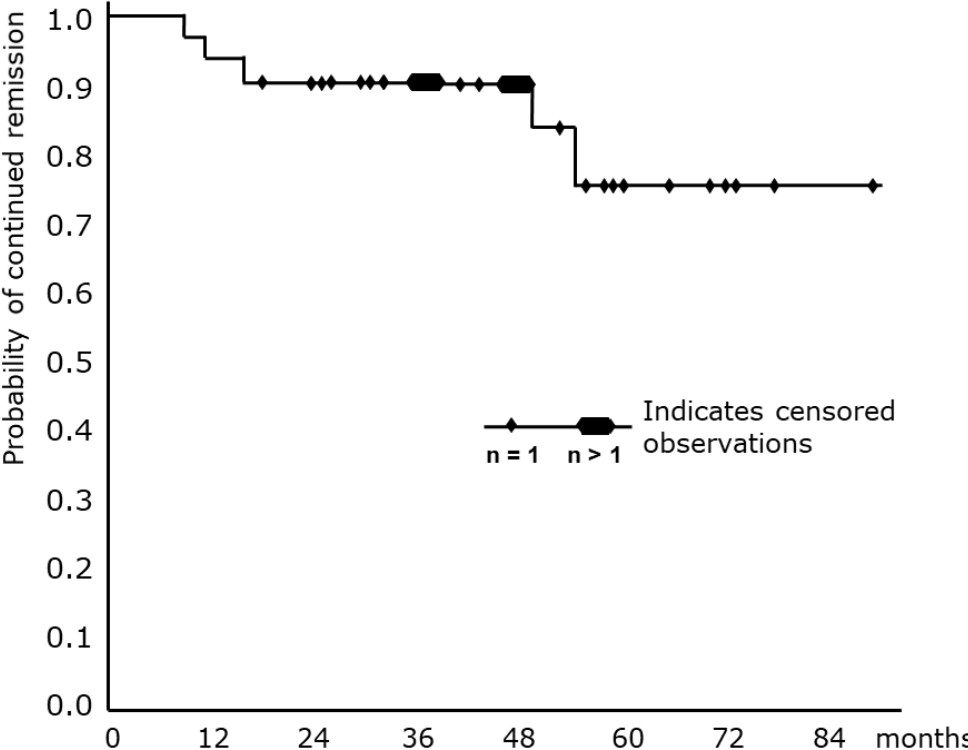

**Figure 5.** Probability of sustained remission in patients who have responded to 4 cycles of rituximab plus bendamustine. Kaplan–Meier plot. Median response duration is not reached after 88 months and the estimated 5-year sustained response rate is 77%. Figure first published in Blood 2020 [26], reused under general permission. © the American Society of Hematology.

Very late responses and deepening of responses over time are also seen after chemoimmunotherapy for WM and are best explained by the existence of long-lived, non-proliferating plasma cells that are more-or-less resistant to therapy [18,22–24,114]. These observations suggest that, even in clinical practice, treatment should be discontinued after 4 cycles irrespective of remission status. Conversely, lack of response after 1–3 cycles should not lead to discontinuation of therapy. The response duration graph (Figure 5) seems to reach a plateau after 50 months, suggesting that a subset of patients will achieve very long-lasting remission, and some may even be cured [26].

Bortezomib monotherapy was studied in a small, non-randomized prospective trial in which 6 of 19 patients responded to one cycle of therapy (3/19 CR, 3/19 PR) [115]. The criteria for CR criteria did not include regression of the bone marrow LPD, which was, however, achieved in most responders. Remission lasted for at least 16 months in

4/6 responders. In theory, these response rates might be improved by using bortezomib in combinations or for an extended duration.

Bruton tyrosine kinase (BTK) inhibitor therapy for CAD has recently been described in a retrospective series of 13 patients with CA-mediated AIHA (including CAD and CAS associated with low-grade lymphoma) who were treated with ibrutinib [116]. All patients responded. The drug was generally well tolerated, but some non-hematologic toxicity was noted. This highly favorable outcome should be confirmed in a prospective trial.

## 10. Complement-Directed Therapies

Therapeutic complement inhibition in CAD was first described in a patient successfully treated with the anti-C5 monoclonal antibody eculizumab [117] and later in a prospective trial comprising 12 patients with CAD and 1 with severe CAS [103]. However, C5 inhibition will not block the C3b-mediated, extravascular hemolysis and, not unexpectedly, the prospective trial showed efficient blockade of C5 but only a small increase in Hb [103].

Classical pathway inhibition would be expected to do better [20,81,118,119], although the results of a small prospective trial of plasma-derived C1 inhibitor in AIHA were disappointing [120]. However, the mouse monoclonal anti-C1s antibody TNT003 was shown to efficiently block classical complement pathway activation and opsonization of erythrocytes with C3b in an in vitro system with patient plasma as a source of CA and normal human serum as a source of complement [79]. These findings led to the development of the corresponding humanized monoclonal IgG4 antibody, sutimlimab (TNT009, BIVV009) (Figure 6) [95,96,121,122].

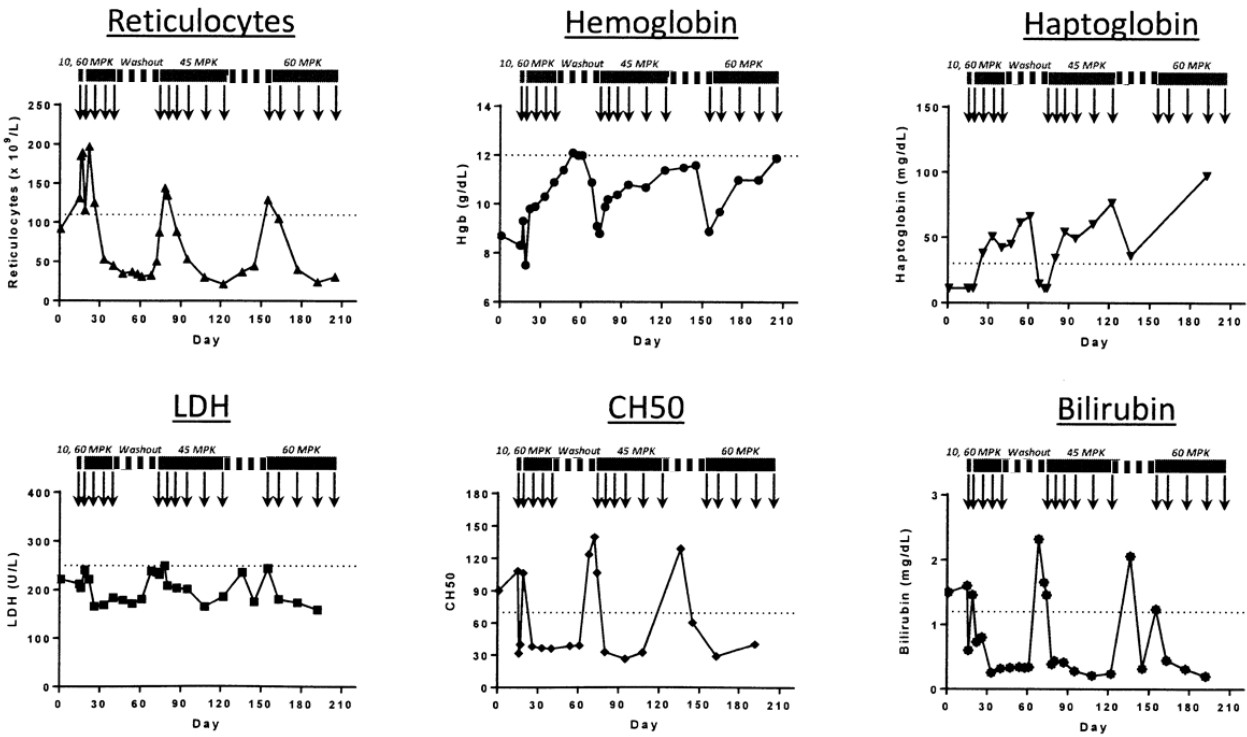

**Figure 6.** Changes in hemoglobin levels, markers of hemolysis, and complement activation assay (CH50) in a representative patient with CAD treated with sutimlimab in a phase 1b trial. All variables normalized rapidly during treatment periods and resumed abnormal levels at the end of 30 days washout, but normalized rapidly again on re-treatment. Continuous horizontal bars, treatment periods; arrows, administration of sutimlimab; hatched horizontal bars, washout periods. LDH, lactate dehydrogenase; MPK, mg per kg body weight. Reproduced from Jäger, D'Sa, Schörgenhofer et al. Blood 2019 (Supplemental data) [122] under general permission. © the American Society of Hematology.

In a nonrandomized phase 2–3 trial (the CARDINAL study), 24 patients with CAD, Hb < 10 g/dL at inclusion, and a recent history of transfusion received sutimlimab at a fixed, bi-weekly dose of 6.5 g in patients weighing < 75 kg and 7.5 g in those weighing ≥75 kg [96,123]. Patients were vaccinated against meningococci, pneumococci, and *Haemophilus influenzae* B, but routine antimicrobial drug prophylaxis was not encouraged. Thirteen patients (54%) met the criteria for the primary endpoint (Hb level > 12.0 g/dL or an increase in Hb of > 2.0 g/dL without transfusion), and 20 patients (83%) achieved a stable improvement in Hb by > 1.0 g/dL. Hb levels increased significantly during the first week; the mean Hb level in the whole cohort exceeded 11.0 g/dL from week 4. Bilirubin levels declined immediately and normalized by week 3. A meaningful reduction in FACIT fatigue score was noted by week 1 and was maintained throughout the study [96]. Adverse events (AEs) with unconfirmed association to the drug were noted in 22 patients (92%); the most common AEs were respiratory tract infection, viral infection, diarrhea, dyspepsia, cough, arthralgia, arthritis, and edema. None of these complaints led to discontinuation of therapy. Meningococcal infections did not occur. The effect and tolerability were unchanged after 1 year on therapy [123]. These favorable results were confirmed in the CADENZA study, a randomized, placebo-controlled, double-blind phase 3 trial of 42 transfusion-independent CAD patients with Hb < 10 g/dL [99].

In a follow-up study of 6 patients who had responded to sutimlimab, discontinuation was followed by immediate relapse, but complement inhibition and control of hemolysis were rapidly restored when treatment was restarted [124]. A recent small prospective trial investigated response duration in 4 patients who had already been successfully treated with sutimlimab [125]. Treatment was discontinued for drug-unrelated reasons in 1 patient, but 3 patients continued for a mean duration of 52 months, including previous sutimlimab therapy. The effects on Hb and bilirubin levels were sustained throughout the study. According to a case report, sutimlimab prevented complement enhancement and exacerbation of CAD in a patient who underwent cardiac surgery [93]. As expected, the sutimlimab studies did not show any improvement of cold-induced circulatory symptoms, as these symptoms are not complement-mediated [95,96,99].

C3 inhibition is also a promising approach. The pegylated cyclic peptide pegcetacoplan (APL-2), a compstatin analog that inhibits C3 and C3b, has shown an excellent clinical effect in paroxysmal nocturnal hemoglobinuria by blocking the whole complement system when administered by subcutaneous infusion [126,127]. A phase 2 trial showed efficient complement blockade with rapid improvement of hemolysis and anemia even in CAD [127]. Patients were vaccinated as in the C1s inhibitor trials. Pegcetacoplan is now being studied in a phase 3 trial in patients with CAD (NCT05096403).

As patients with >10% infiltration with clonal lymphocytes in the bone marrow were excluded from the CARDINAL and CADENZA studies, the effect of C1s inhibition in patients with a more prominent lymphoid infiltration and/or high IgM levels has not been sufficiently explored [95,96,99]. In theory, upstream complement blockade might carry a risk of severe bacterial infection. Thus far, studies indicate that C1s and C3 inhibition are safe in this regard provided the patients are properly vaccinated against encapsulated bacteria [96,99,128]. As hereditary deficiencies in proximal classical pathway components are associated with systemic lupus erythematosus (SLE), a risk of developing SLE might be suspected when these components are blocked [129]. Clinical data have not supported this concern [96,99,122,123].

## 11. Supportive Measures

### 11.1. Transfusion

When indicated, patients with CAD can be safely transfused provided that specific precautions are observed, different from those required in warm-AIHA. It is usually possible to find compatible erythrocytes by crossmatching at 37 °C, although at high thermal amplitude, absorption techniques are sometimes also necessary. However, in vivo cooling during transfusion can cause agglutination and hemolysis of patient as well as donor

erythrocytes. The patient and the extremity chosen for infusion should be kept warm, and most authors recommend the use of an in-line blood warmer [1,5,84].

### 11.2. Plasmapheresis

Plasma exchange has a strong theoretical rationale in CAD because 80% of IgM is found intravascularly [130]. There are several reports on immediate and favorable effect, but unsuccessful attempts are probably underreported [5,106,131]. Today, plasma exchange is only recommended in critical situations [5]. Because exogenous complement may exacerbate hemolysis, albumin should be used instead of plasma for substitution [84,85]. The effect is short-lived, and pharmacological therapy should be initiated concomitantly [5,84,132].

### 11.3. Folic Acid

The clinical evidence for folic acid supplementation is an old study of sickle cell anemia [133], whereas serum folate levels were normal in 97% of patients with CAD in a large study, even before taking supplements [26]. Still, folic acid supplementation (1–5 mg/day) is usually recommended because of consumption due to hemolysis and bone marrow compensation [5].

### 11.4. Erythropoietin

High doses of erythropoietin or its analogs has been shown to moderately increase Hb levels in patients with warm as well as cold-reactive AIHA and can be used as a supplement in the second or third line, in the setting of fulminant presentation, or as a "bridging" to the often slow-acting B-cell-directed therapies [134,135].

### 11.5. Thrombosis Prophylaxis

There are no hard data regarding antithrombotic prophylaxis in CAD. In the absence of evidence-based guidelines, we recommend administration of low-molecular-weight heparin in prophylactic doses or a direct oral anticoagulant in hospitalized patients with severe anemia and patients with acute exacerbations or additional risk factors [1,5,6].

## 12. Choice of Therapy

Ideally, preferences regarding the choice of B-cell-directed versus complement-directed therapies should be clarified by comparative studies, but it remains doubtful whether such trials will ever be undertaken in this rare disease. B-cell-directed therapy may still be preferred in the first line if not contraindicated and, provided the frequently long time to response, is deemed to be acceptable [1,5,95]. Rituximab plus bendamustine should be considered in the first line for otherwise fit patients with severe CAD, while significant comorbidity will favor rituximab monotherapy [5,111,113]. B-cell-directed therapy should also be preferred in patients with disabling cold-induced circulatory symptoms and, until further evidence exists, in those with lymphoid bone marrow infiltration > 10% [95]. For second-line B-cell-directed therapy, ibrutinib or bortezomib can be considered.

In the second line, complement-directed therapy with sutimlimab will often be the best choice if available and affordable [95,96,99]. Sutimlimab should be considered even in the first line if a rapid response is required, for example, in severely anemic patients and those with acute exacerbations that do not resolve spontaneously [1,95,96,99]. Sutimlimab may also be used as a "bridging" to the more slow-acting B-cell-directed approaches [1,94,95]. Patient preferences should be emphasized.

## 13. The Future

Following the highly promising results of the retrospective ibrutinib study [116], therapy with a BTK inhibitor should be further explored in a prospective trail. Furthermore, the anti-CD38 monoclonal antibody daratumumab has been reported to induce remission in 2 patients in whom other chemoimmunotherapy had failed [24,136], and a systematic

series would be of interest for possible treatment in the second or subsequent line in highly selected patients.

Complement-directed therapies should also be further studied. Among these, the phase 3 trial of pegcetacoplan is of high interest (NCT05096403) [127]. Other candidate drugs include the C1q inhibitor ANX005 [137,138], the C1s inhibitor BIVV020 (NCT04802057) [81], and the C2 inhibitor ARGX-117 [139], none of which have been evaluated in published clinical trials.

## 14. Conclusions

CAD is a type of AIHA and a distinct clonal LPD, in most cases not overlapping with WM. In atypical cases of *MYD88* L265P-positive WM with cold agglutinin-mediated AIHA, it might be a matter of discussion whether these should be classified as CAD or CAS. Patients should be offered treatment if they have symptomatic anemia, severe fatigue, or cold-induced circulatory symptoms interfering with daily living. Effective treatments are directed against the pathogenic B-cell clone or the classic complement pathway, and the choice of therapy should be made on an individual basis. Additional therapeutic approaches are in the pipeline, for example, BTK inhibition and further development of complement-directed therapies. Therefore, patients with CAD requiring treatment should be considered for prospective trials.

**Funding:** This research received no external funding.

**Institutional Review Board Statement:** Not applicable.

**Informed Consent Statement:** Not applicable.

**Data Availability Statement:** Not applicable.

**Conflicts of Interest:** The authors declare no conflict of interest related to the preparation of this article. Outside this work, S. Berentsen has received advisory board honoraria from Anexon, Momenta Pharmaceuticals, Sanofi-Genzyme, and Sobi and lecture honoraria from Apellis, Janssen-Cilag, Sanofi-Genzyme, and Sobi. S. D'Sa has received advisory board honoraria from Sanofi-Genzyme; coverage of meeting expenses from Sanofi-Genzyme, BeiGene, and Janssen-Cilag; and research funding from Janssen-Cilag and BeiGene. U. Randen and A. Małecka report no conflicts of interest. J.M.I. Vos has received advisory board honoraria and consultancy fees from Sanofi-Genzyme, research support from BeiGene (all institutional), and travel support from Celgene.

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
