# Peer review of "Cold Agglutinin Disease: Improved Understanding of Pathogenesis Helps Define Targets for Therapy"

_hemato, doi:10.3390/hemato3040040_

Round 1

Reviewer 1 Report

This is a very well written, detailed, complete and comprehensive review about cold agglutinine disease. The Authors deal with all aspects concerning the pathology. In my opinion, no changes are required.

Author Response

Thank you for this highly favorable evaluation.

Reviewer 2 Report

The paper is technically sound and provides further support to the concept that Cold Agglutinin Disease (CAD) is a type of AIHA and a distinct clonal LPD, in most cases not overlapping with WM. This is a well updated review that will of help in the differential diagnosis of CAD.

The authors discuss if atypical cases of MYD88 L265P positive WM with cold agglutinin-mediated AIHA should be classified as CAD . ON the basis of pathophysiological issues, interesting aspects of treatment are discussed and new therapeutic challenges are proposed

Author Response

(The authors gave the same response as above.)

Reviewer 3 Report

Very nice review of the CAD. I have minor revisions.

Line 123. I think that you might consider to add the International Consensus Classification, with the paper published in Blood (Campo et al.).

Line 144. It will be important to add a comment regarding the possible relation with marginal zone lymphoma due to the presence of the same cytogenetical and mutational alterations.

Line 154. The pattern of infiltration of the bone marrow is nodular and interstitial. This is not a diffuse pattern, in my opinion.

Author Response

Thank you for this highly favorable evaluation. Our response to your few, minor comments is as follows,

Line 123. I think that you might consider to add the International Consensus Classification, with the paper published in Blood (Campo et al.).

We fully agree. In the revised version, we have added the International Consensus Classification in the same sentence. The Campo et al paper in Blood has now been referenced by replacing reference #62 (a secondary article that did not add anything substantial to reference #61) with the article by Campo et al.

Line 144. It will be important to add a comment regarding the possible relation with marginal zone lymphoma due to the presence of the same cytogenetical and mutational alterations.

            We agree. However, a sentence in Section 4 (CAD and Waldenström’s macroglobulinemia: distinction or overlap?), line 186-7 in the first version, already points out the similarities with MZL, and we do not find it necessary to repeat this statement.

Line 154. The pattern of infiltration of the bone marrow is nodular and interstitial. This is not a diffuse pattern, in my opinion.

            We have replaced the word diffuse by interstitial.